# FinTech Companies: A Bibliometric Analysis †

**Gencay Tepe** [1,*], **Umut Burak Geyikci** [1] and **Fatih Mehmet Sancak** [2]

1 Faculty of Business, Manisa Celal Bayar University, 45140 Manisa, Turkey; umutburak.geyikci@cbu.edu.tr
2 Dikkan Group, 35730 Izmir, Turkey; fatih.sancak@dikkan.com
\* Correspondence: gencay.tepe@cbu.edu.tr; Tel.: +90-236-201-39-25
† This paper was presented as an oral presentation at the 9th World Conference of Business Economics Management, Porto, Portugal, 1–3 October 2020.

**Abstract:** The financial-technology industry has recently attracted the attention of many sectors. The financial-technology industry designs new and unusual technological financial services in many areas. It combines technology with finance and provides an alternative to the traditional financial system. In the scope of this study, 636 publications were obtained from Scopus. Various tools, such as Microsoft Excel for frequency analysis, and VOSviewer for data visualization, were used. The open-source codes used for bibliometric analysis through the R Studio program were developed by the authors and used for citation-metrics analysis. The main aim of this study was to find out the most influential studies and authors and to reveal the distributions and impacts of publications in the FinTech area between 2015 and 2021 from the Scopus database. The results indicate that the most influential journal is Sustainability Switzerland, and the most cited author is Gomber et al. Additionally, Rabbani has the most publications, while China has emerged as the most productive country. On the other hand, this study found that FinTech research clustered in four areas. These areas are computer science, business management, economics, and social sciences. This FinTech study examines financial services, financial access, and financial technology, where FinTech is at the center. It also focuses on cryptocurrency, bitcoin, and smart contracts where the blockchain is at the center. The results reveal a systematic map of existing studies. Further, the study plays a guiding role in future research.

**Keywords:** FinTech; financial technology; blockchain; financial information systems

## 1. Introduction

Although it only entered the literature five years ago, FinTech has been studied a lot. It refers to companies and Finance 4.0 that create financial technologies at the highest level. Globally, FinTech is being implemented rapidly in human life in recent years.

Of recent FinTech studies, some focus on all aspects of the issue in general (e.g., Arner et al. 2016; Zalan and Toufaily 2017; Dospinescu et al. 2021), while others examine more-specific aspects. These include studies related to banks and traditional financial institutions (Kotarba 2016; Buchak et al. 2018; Hu et al. 2019), venture capital, cryptocurrencies, and blockchain (Kaplan and Lerner 2016; Ante et al. 2018; Gozman and Willcocks 2019; Kim et al. 2018; Ji and Tia 2021; Mora et al. 2021), insurance (Yan et al. 2018b; Stoeckli et al. 2018), and asset management (Rogowski 2017; Dugast and Foucault 2018). While each study adds an important perspective on the subject, a bibliometric analysis can provide a broader perspective and assessment than has been the case for studies thus far.

A network analysis carried out through bibliometric analysis defines new areas and information on the subject more strongly. It can also identify research groups and researchers to show how various areas of thought have emerged. Finally, it can identify leading and influential researchers in these research groups, identify different and new issues addressed by these influential researchers, and identify areas of study related to these new issues.

This study provides a detailed and comprehensive analysis by identifying researchers and publications with high influence in this pool, starting with 401 studies focusing on future-oriented FinTech applications. Various performance indicators were calculated for the bibliometric analysis. The formulization of the methods used was encoded by the authors on an R-based basis using the R Studio 1.2 program. The data processed through the program were then handled with Gephi 0.8.2 and VOSviewer 1.6.11 programs for visualization and mapping purposes to obtain the final outputs.

FinTech, which is short for financial technology, has spread rapidly worldwide, although its importance varies from country to country depending on the level of economic development and market structure (Berkmen et al. 2019). The concept, which originated in the early 1990s, currently refers to a rapidly evolving process in financial services (Arner et al. 2017; Hochstein 2015). FinTech describes companies offering financial services using modern creative technologies that *"attract customers with products and services that are more user-friendly, efficient, transparent and automated than those currently available"* (Dorfleitner et al. 2017, p. 5). FinTech firms cannot be defined within legal parameters because they operate in different business lines and models, and a wide range of industries, from crowdfunding to credit providers, cryptocurrencies to angel-investment networks.

FinTech has developed through three basic stages (Arner et al. 2017). The first phase resulted from surplus production and technological innovations brought about by the industrial revolution with the use of the first simple abacuses. After the mid-1800s, the invention of the telegraph (Nicoletti 2017), and telegraph communication and intensive trade between countries, enabled financial transactions to be made on a global scale using technology (Standage 2013). From 1866 to 1967, the financial services industry was heavily connected with technology but remained largely analog. This period is called FinTech 1.0.

Developments in digital technology between 1967 and 2008, known as FinTech 2.0, enabled financial-services technologies to switch from analog to digital and become globalized. For example, Barclay's Bank was the first to introduce automatic teller machines (ATMs) in 1967 (Nicoletti 2017), while electronic payment systems significantly changed the financial structure, making money transfers between banks and countries quite easy. In 1975, the Basel Committee was established to reduce the risks of interbank money transfers, with new rules to regulate relations between international banks (History of the Basel Committee 2019).

In terms of what consumers expect from a bank, e-banking/m-banking, the possibility of performing ATM cash transactions, customer service, ease of use, and the volume of information on the card have become very important (Dospinescu et al. 2019). On the other hand, rapidly developing financial integration that connected global markets marked a new era, FinTech 3.0, after 2008, especially with the introduction of the Internet, when Wells Fargo presented the first internet-banking experience, and startups and technology companies began offering financial products and services directly to businesses and consumers (Arner et al. 2016). These FinTechs considerably damaged the profitability of the banking sector (Zalan and Toufaily 2017).

While we currently still live in FinTech 3.0, a future upgrade to FinTech 4.0 will happen. Arner et al. (2017) even claim that FinTech 4.0 arrived in 2018, thanks to applications such as the Internet of things, big data, artificial intelligence, and cloud computing.

Figure 1 shows how FinTech is segmented into four fundamental sectors (Dorfleitner et al. 2017). Financing is the segment that provides funding for individuals or organizations through crowdfunding and credit and factoring. Crowdfunding usually involves raising small amounts of money from large numbers of people via the Internet or social media. The most important feature is the setting of the deadline. If the target amount cannot be reached within the specified period, the operation is canceled (Lee and Kim 2015). Credit and factoring are the processes by which FinTech firms provide financing to individuals or companies cheaply and quickly by automating transactions in collaboration with banks. The second main sector, asset management, includes services such as social trading, Robo-advice, personal financial management (PFM), and investing and banking. The third sector,

payment, refers to national and international payment transactions. These include virtual payment methods, such as cryptocurrencies and blockchains, which are used as alternatives to conventional monetary transactions. Finally, other FinTechs cannot be classified within the first three traditional banking functions. These include insurance, search engines, comparison sites, technology, IT, and infrastructure.

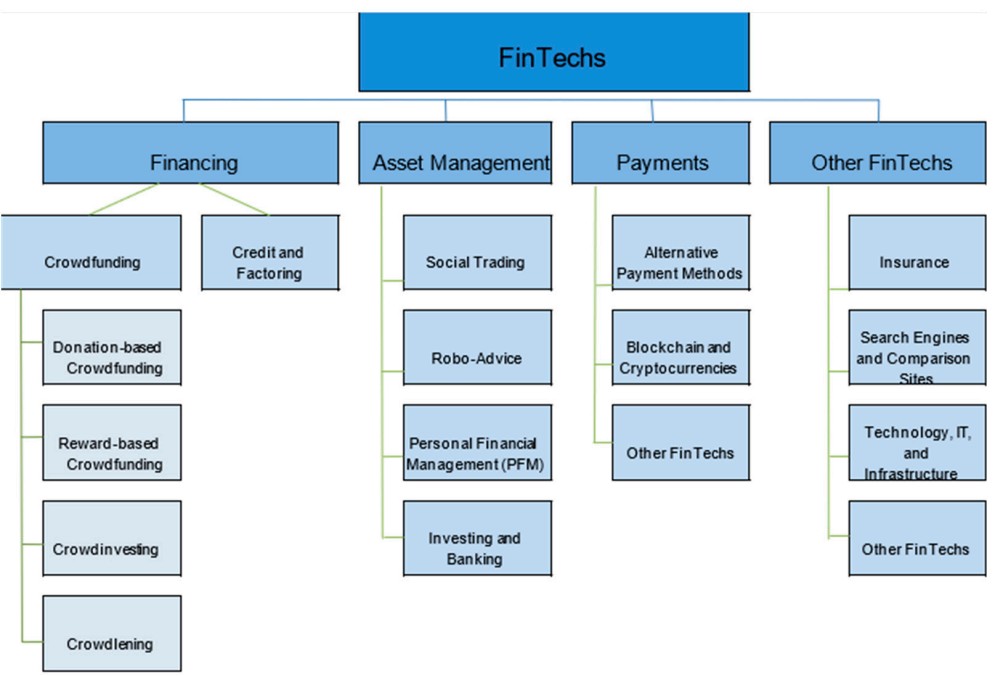

**Figure 1.** FinTech Segmentations. **Source:** Dorfleitner et al. (2017), cited by Al-Ajlouni and Al-Hakim (2018).

The term bibliometric was first used in the Journal of Documentation (Fairthorne 1969). Bibliometrics (sometimes called scientometrics) involves quantitative analysis, which is the main tool of science. It provides a statistical analysis of data, such as how frequently journal articles are cited. Comparisons can be made across countries and research branches. While bibliometric analyses are becoming increasingly popular, their novelty means that there are no studies yet that directly investigate FinTech. Only two (Milian et al. 2019; Wu 2017) have used the word FinTech in their titles. However, they did not exclusively focus on it. Instead, their analyses drew on the following segments: payments, deposit, and landing, insurance, capital raising, investment management, and market provisioning. A few bibliometric studies have focused on individual segments within FinTech, such as crowdfunding (Martínez-Climent et al. 2018; Blasco-Carreras et al. 2015; Blažun Vošner et al. 2017), payments (Karafiloski and Mishev 2017; Cao et al. 2017; Dabbagh et al. 2019; Zheng et al. 2018; Merediz-Solà and Bariviera 2019; Liu 2016), asset management (Yan et al. 2018a), and other financing functions (Kumari and Sharma 2017; Cancino et al. 2017).

## 2. Research Method

The basic aim of a bibliometric analysis is to collect previous literature and related topics on the research subject to form objective findings that can be tested and replicated. It aims to both categorize previous studies and offer a rigorous methodological examination of the research results. To show that the study adds new information to the literature, the results should be defined in accordance with the research questions.

*Research Questions*

In the present study, FinTech-related publications and researchers were subjected to structural categorical analysis. Following Milian et al. (2019), the following two ba-

sic research questions with three sub-questions and two others, a total of seven, were addressed.

RQ1. How has the literature developed between 2015 and 2021?

RQ1.1. What are the most influential studies and authors?

RQ1.2. What are the main studies in FinTech?

RQ1.3. What are the distributions and impacts of publications over time?

To respond to RQ1, it is necessary to group the important studies, identify the relationships between them, and categorize them within the framework of current studies. This leads to the second question:

RQ2. What are the important topics in the FinTech literature?

In responding to this question, Lotka's Law and Bradford's Law, which are classics in bibliometric analysis, were assessed for their compatibility with the data.

RQ3. Are the results compatible with Lotka's Law?

RQ4. Are the results compatible with Bradford's Law?

## 3. Sampling and Methodology

Bibliometric analysis was used to determine the scope of the scientific FinTech literature. The bibliometric analysis used in this research is a very detailed and comprehensive analysis technique in this field.

*Bibliometric Analysis*

The bibliometric analysis identifies the most prolific countries and universities, and the most influential authors, studies, and journals. The FinTech and bibliometric analysis dataset for the study was taken from Scopus. While Web of Science was also scanned, it was excluded from the evaluation because it provided considerably fewer studies than Scopus. The bibliometric analysis technique aimed to reveal the evolution of the FinTech research literature in terms of RQ1.1, RQ1.2, and RQ1.3, specifically the most influential articles, authors, and topics.

Many factors can be examined in bibliometric analysis. However, the analysis to be performed must be suitable for the purpose. The present study followed the method proposed by Cadavid Higuita et al. (2012), Albort-Morant and Ribeiro-Soriano (2016), and Martínez-Climent et al. (2018). In this method, the indicators are divided into three types: quantity, quality, and structural indicators (Martínez-Climent et al. 2018). (1) Quantity indicators contain numerical data for the area to be analyzed. (2) Quality indicators show the academic impact of publications. (3) Structural indicators reveal the relationships between publications.

Social network analysis is used for measuring both quality and structural indicators. In social network analysis, the network consists of nodes connected through networks (Wasserman and Faust 1994). This determines the centrality of each author by the number of connections they make with other members of the network. Centrality has three main principles: degree, closeness, and betweenness (Freeman 1979, cited by Milian et al. 2019). Centrality degree indicates how many co-publications an author has. Betweenness measures the number of times a node captures the shortest route between two other nodes, and thus shows the binding role that the author plays among other authors. Farness is the sum of the shortest distance of one node from other nodes while proximity is the opposite of farness. The greater the degree, the less the total distance from one node to all other nodes (Milian et al. 2019). Authors with high proximity reach new information faster and spread their ideas more quickly.

RQ2 addresses the issue of which topics FinTech researchers focus on while Lotka's Law (Lotka 1926), which measures authors' scientific productivity, was addressed through RQ3. According to Lotka's Law, the number of authors contributing to the literature with n number of studies is $1/n^2$ of the number of authors contributing to the literature with a single study.

RQ4 assesses Bradford's Law (Bradford [1929] 1985), which determines the distribution of references to journals. According to this law, a bibliographic study on any subject will show that there is a small core group of journals that publishes a third of all articles in this field. A second, larger group of journals publishes the next third while the biggest group of journals publishes the remainder.

## 4. Findings

This section presents the results, periods, publications, authors, and other information of the analyses.

A total of 636 publications were scanned in Scopus for academic papers on FinTech (including journal articles, conference papers, books, and book chapters). Figure 2 specifies the number of publications found by years.

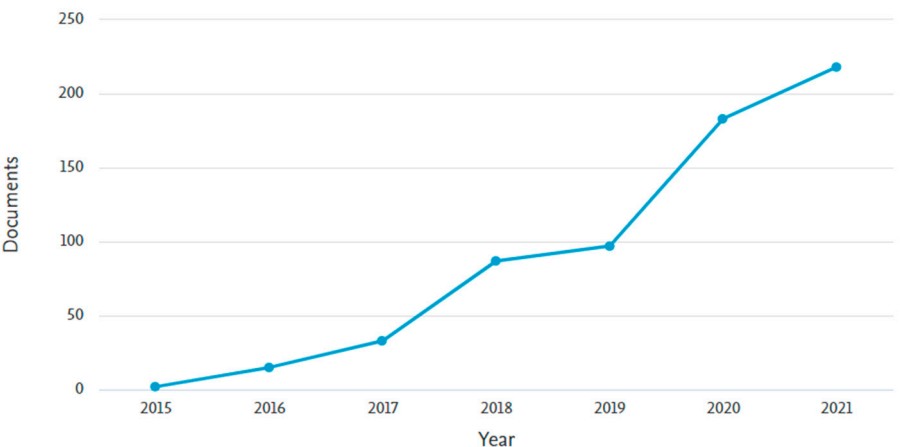

**Figure 2.** Publishing trend in FinTech. Note(s): This figure represents the publication trend of academic papers on FinTech between 2015 and 2021. The data were retrieved from the Scopus database using the keyword "FinTech".

Figure 2 shows that FinTech has grown geometrically since 2015 when it first emerged as a concept. The papers were written by 1445 different authors from 387 different journals and books. In the scanned sources, the average citations per document were 7.52, the number of documents per author was 0.44 while the collaboration index was 2.75.

Table 1 shows which ten universities had the most affiliations of FinTech authors. Universities in Asian countries have contributed the most (6 of the top 10). Table 1 also shows total production (TP), total citations (TC), and citations per publication (CPP).

**Table 1.** Top 10 university affiliations by documents.

| No | University | TP | TC | CPP | h-Index |
|----|------------|----|----|-----|---------|
| 1 | Bina Nusantara University | 16 | 30 | 1.88 | 3 |
| 2 | Amity University | 11 | 8 | 0.72 | 2 |
| 3 | The University of Sydney | 10 | 181 | 18.1 | 3 |
| 4 | UNSW Sydney | 9 | 191 | 21.2 | 4 |
| 5 | Ahlia University | 9 | 18 | 2 | 3 |
| 6 | Soongsil University | 8 | 110 | 13.75 | 3 |
| 7 | Universitas Indonesia | 8 | 12 | 1.5 | 2 |
| 8 | Singapore Management University | 7 | 244 | 34.85 | 4 |
| 9 | Universidad Anáhuac México | 7 | 0 | 0 | 0 |
| 10 | Kingdom University | 6 | 55 | 9.17 | 4 |

Note(s): This table was created with a dataset from Scopus via Excel.

Figure 3 shows which 10 institutions sponsored the most FinTech papers.

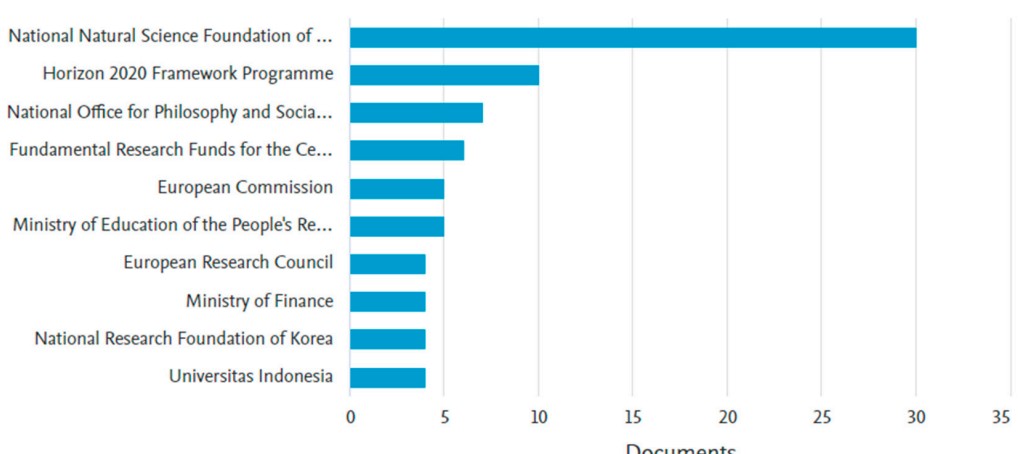

**Figure 3.** Top 10 funding sponsors of documents. Note(s): This figure represents the 10 institutes that sponsored the most academic articles on FinTech between 2015 and 2021. The data were taken from the Scopus database using the keyword "FinTech".

Figure 4 shows the geographical locations of all contributing countries, with the number of publications decreasing from dark to light blue, while grey indicates no contribution. China, the USA, and the UK have the highest contributions.

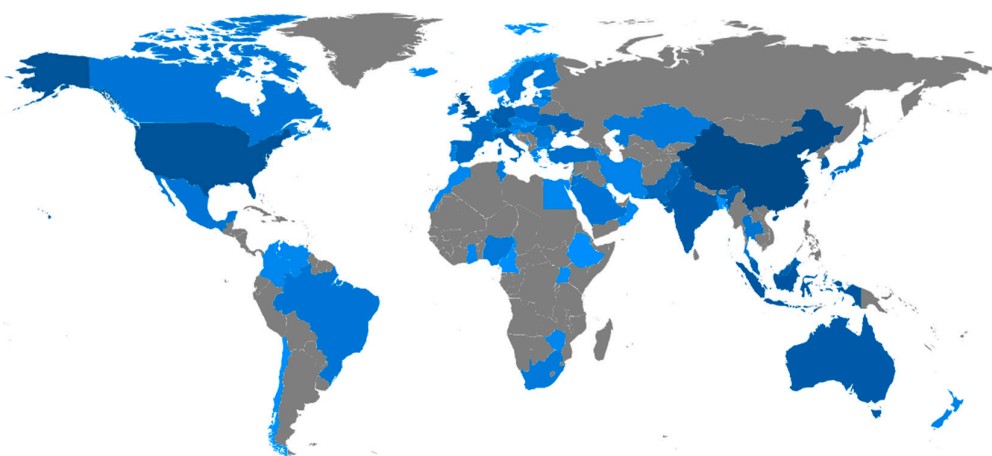

**Figure 4.** Geographical locations of contributing countries. Note(s): This figure was created with a dataset from Scopus via R Studio.

Figure 5 shows the country collaboration map. UK authors have 59 joint publications with authors in other countries, including 7 with Chinese authors, 6 with Australian authors, and the remaining 46 collaborations with authors in 25 different countries. The UK is followed by the US with 54 collaborations, China with 52, Australia with 43, and Singapore with 18.

Table 2 lists the 10 most productive journals. The journals that are not in the field of Finance and Entrepreneurship were excluded from the analysis results. At the top, Sustainability Switzerland has the most publications on FinTech, and a TC value of 96, whereas the second-ranked, Lecture Notes in Computer Science including subseries, has a TC value of 40. Despite only ranking tenth, Industrial Management and Data Systems has the highest CPP value at 33.2, and Financial Innovation has the second-highest CPP value at 15.28.

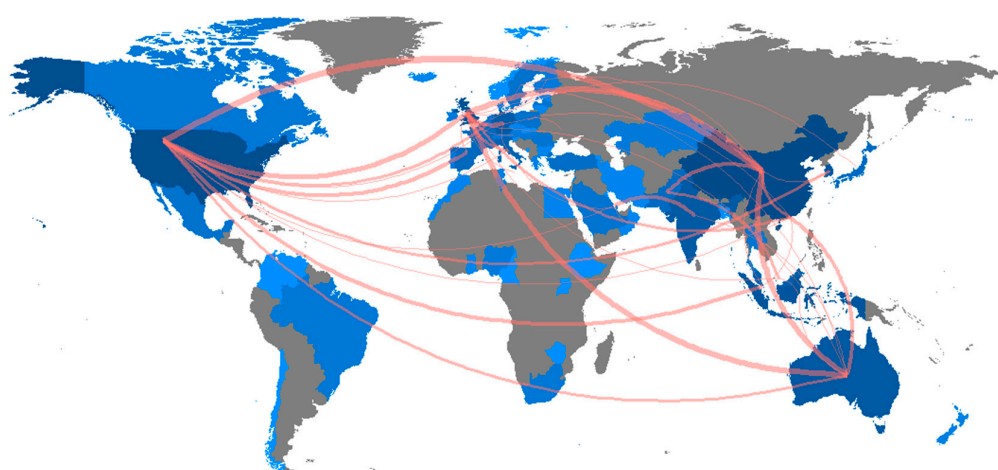

**Figure 5.** Country collaboration map. Note(s): This figure was created with a dataset from Scopus via R Studio.

**Table 2.** Most-productive journals.

| No | Journals | TP | TC | CPP | h-İndex |
|----|----------|----|----|-----|---------|
| 1 | Sustainability Switzerland | 15 | 96 | 6.4 | 5 |
| 2 | Lecture Notes in Computer Science including subseries | 8 | 40 | 5 | 1 |
| 3 | Journal of Open Innovation: Technology, Market, and Complexity | 8 | 33 | 4.12 | 3 |
| 4 | Finance Research Letters | 8 | 32 | 4 | 2 |
| 5 | Perspectives In Law, Business and Innovation | 8 | 0 | 0 | 0 |
| 6 | Financial Innovation | 7 | 107 | 15.28 | 4 |
| 7 | Impact Of Financial Technology (Fintech) on Islamic Finance and Financial Stability | 7 | 6 | 0.86 | 2 |
| 8 | Palgrave Studies in Democracy, Innovation, and Entrepreneurship for Growth | 7 | 0 | 0 | 0 |
| 9 | ACM International Conference Proceeding Series | 6 | 20 | 3.3 | 3 |
| 10 | Industrial Management and Data Systems | 5 | 166 | 33.2 | 4 |

Note(s): This table was created with a dataset from Scopus via Excel.

As the most productive country, China has 87 publications, 745 citations, and 8.56 citations per publication. Six of the top ten are Asian countries, two are European, while Australia represents Oceania. The most productive countries all have h-index values of 4 or above (Table 3).

**Table 3.** Most-productive countries.

| No | Countries | TP | TC | CPP | h-Index |
|----|-----------|----|----|-----|---------|
| 1 | China | 87 | 745 | 8.56 | 13 |
| 2 | United States | 84 | 1415 | 16.84 | 18 |
| 3 | United Kingdom | 67 | 615 | 9.18 | 12 |
| 4 | Indonesia | 49 | 137 | 2.79 | 7 |
| 5 | South Korea | 41 | 673 | 16.41 | 12 |
| 6 | Australia | 41 | 416 | 10.14 | 8 |
| 7 | India | 36 | 57 | 1.58 | 4 |
| 8 | Germany | 31 | 762 | 24.58 | 10 |
| 9 | Malaysia | 25 | 87 | 3.48 | 6 |
| 10 | Singapore | 21 | 287 | 13.6 | 7 |

Note(s): This table was created with a dataset from Scopus via Excel.

### 4.1. Author Influence

Table 4 shows which authors have been most prolific. Authors with four or more publications between 2015 and 2021 are listed in Table 4. Rabbani, with the most publications, has a TC score of 55. Tan, who is in second place, has a TC score of 105. More than half of Rabbani and Khan's citations are self-citations. Muthukannan has the lowest CPP score, which indicates a weak correlation between that author's large number of publications and their impact factor.

**Table 4.** The top 10 contributing authors' number of published articles and self-citations.

| No | Authors | TP | TC | CPP | h-Index | Self-Citations |
|----|---------|----|----|-----|---------|----------------|
| 1 | Rabbani, M.R. | 6 | 55 | 9.16 | 4 | 26 |
| 2 | Tan, B. | 5 | 105 | 21 | 2 | 9 |
| 3 | Wójcik, D. | 5 | 19 | 3.8 | 3 | 5 |
| 4 | Muthukannan, P. | 5 | 8 | 1.6 | 2 | 1 |
| 5 | Hornuf, L. | 4 | 156 | 39 | 3 | 9 |
| 6 | Jagtiani, J. | 4 | 89 | 22.25 | 3 | 3 |
| 7 | Gozman, D. | 4 | 65 | 16.25 | 2 | 6 |
| 8 | Wonglimpiyarat, J. | 4 | 52 | 13 | 3 | 0 |
| 9 | Khan, S. | 4 | 42 | 10.5 | 3 | 22 |
| 10 | Ashta, A. | 4 | 39 | 9.75 | 3 | 9 |

Note(s): This table was created with a dataset from Scopus via Excel.

Table 5 shows which authors received the most citations by year. Table 4 above shows citations based on total publications, whereas Table 5 shows the most citations received by one study. The 10 authors with the most citations had 1342 citations in total, with an average of 134.2 citations each.

**Table 5.** Most-cited authors.

| | Publications/Year | <2017 | % | 2017 | % | 2018 | % | 2019 | % | 2020 | % | 2021 | % | Total |
|---|---|---|---|---|---|---|---|---|---|---|---|---|---|---|
| 1 | Gomber et al. (2018) | 0 | 0% | 0 | 0% | 8 | 3% | 43 | 20% | 62 | 29% | 101 | 47% | **214** |
| 2 | Lee and Shin (2018) | 0 | 0% | 0 | 0% | 7 | 3% | 30 | 14% | 82 | 40% | 85 | 41% | **204** |
| 3 | Gomber et al. (2017) | 0 | 0% | 1 | 5% | 19 | 11% | 31 | 17% | 58 | 33% | 65 | 37% | **174** |
| 4 | Gabor and Brooks (2017) | 0 | 0% | 7 | 4% | 19 | 13% | 28 | 19% | 33 | 22% | 59 | 40% | **146** |
| 5 | Buchak et al. (2018) | 0 | 0% | 0 | 0% | 1 | 1% | 15 | 13% | 30 | 25% | 71 | 60% | **117** |
| 6 | Gai et al. (2018) | 0 | 0% | 0 | 0% | 9 | 8% | 32 | 28% | 32 | 28% | 40 | 35% | **113** |
| 7 | Schueffel (2016) | 0 | 0% | 3 | 3% | 16 | 15% | 16 | 15% | 34 | 33% | 34 | 33% | **103** |
| 8 | Leong et al. (2017) | 0 | 0% | 2 | 2% | 9 | 9% | 21 | 21% | 34 | 35% | 32 | 33% | **98** |
| 9 | Haddad and Hornuf (2019) | 0 | 0% | 0 | 0% | 0 | 0% | 6 | 6% | 34 | 36% | 54 | 57% | **94** |
| 10 | Shim and Shin (2016) | 1 | 1% | 5 | 6% | 10 | 12% | 18 | 22% | 22 | 28% | 24 | 30% | **79** |

Note(s): This table was created with a dataset from Scopus via Excel.

Figure 6 shows the frequency of citations of individual articles. The size of each node indicates the number of citations. The most-cited authors were Gomber et al. (2018), Lee and Shin (2018), Gomber et al. (2017), Gabor and Brooks (2017), Buchak et al. (2018), Gai et al. (2018), Schueffel (2016), Leong et al. (2017), Haddad and Hornuf (2019), and Shim and Shin (2016).

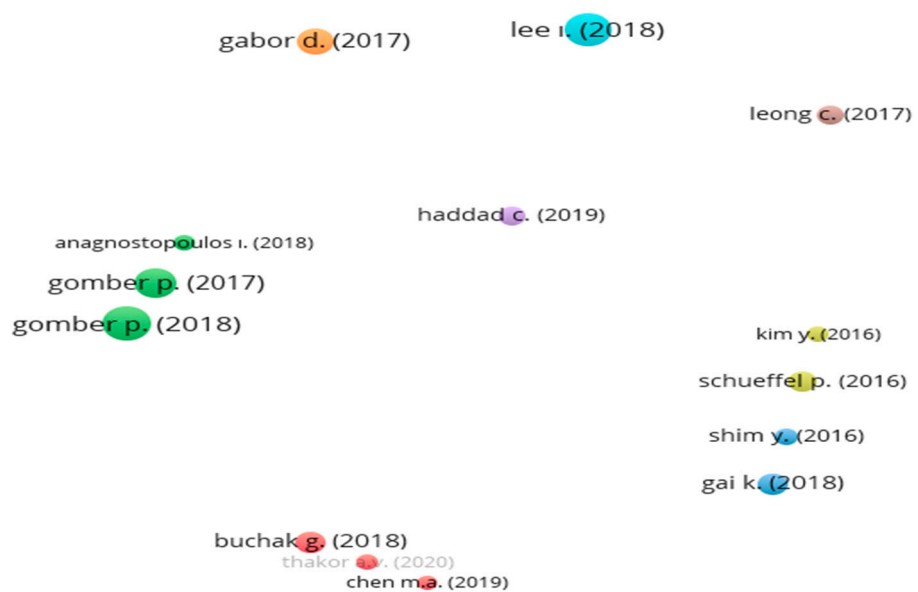

**Figure 6.** Frequency of citations of publications (Fractionalization). Note(s): This figure was created with a dataset from Scopus via VOSviewer.

### 4.2. Centrality of Publications

Bibliometric analysis can also determine the relationship between publications. Node sizes are determined by the number of citations and a network is created by attributing the degree of the node to each citation. The size of the node indicates the degree of centrality. The links show the direction of information flow of direct citations between nodes from the former to the new. Node tags include the degree of total centrality as well as definitions of publications. Figure 7 presents the network structure for nodes with a degree of centrality of more than 10.

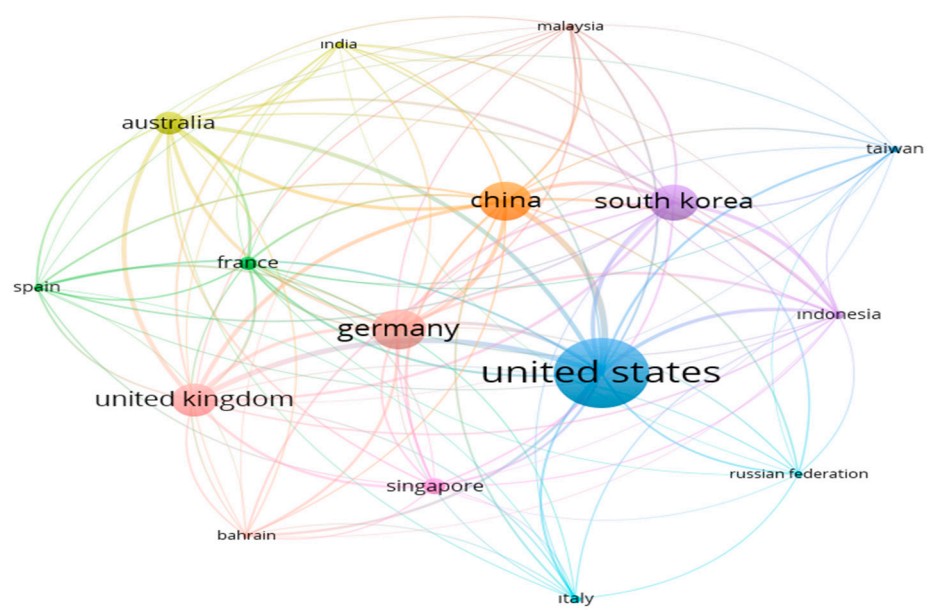

**Figure 7.** Network visualization of the centrality of countries' citations. Note(s): This figure was created with a dataset from Scopus via VOSviewer.

### 4.3. Centrality of Keywords

Figure 8 shows the network of keywords. The lines connecting the nodes represent the relationship between the most commonly used keywords in the articles in the study. These

can be grouped into four sets: financial technology, China, financial services, and blockchain. On the left, crowdfunding, blockchain, and machine learning appear to be grouped around FinTech. The high frequency of these terms shows the interest and up-to-datedness of the researchers.

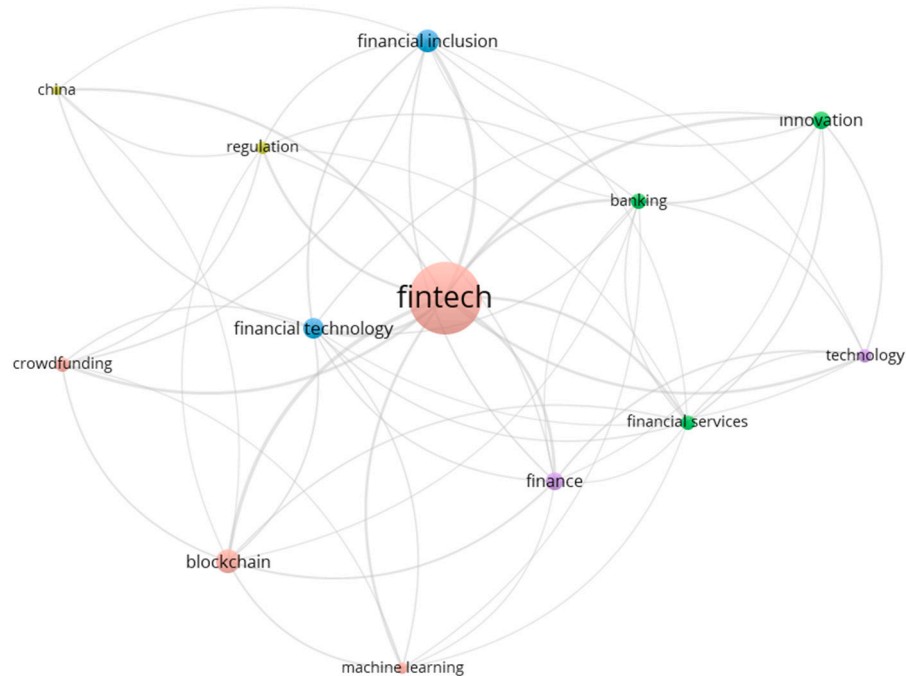

**Figure 8.** Network visualization of the centrality of keywords. Note(s): This figure was created with a dataset from Scopus via VOSviewer.

Regarding the distribution of the scientific fields that the sampled studies come from, business management comes first with 22.7%, followed by computer science with 18.2%, economics (16.7%), and social sciences (13.3%). Thus, a variety of disciplines are conducting research into FinTech (Figure 9).

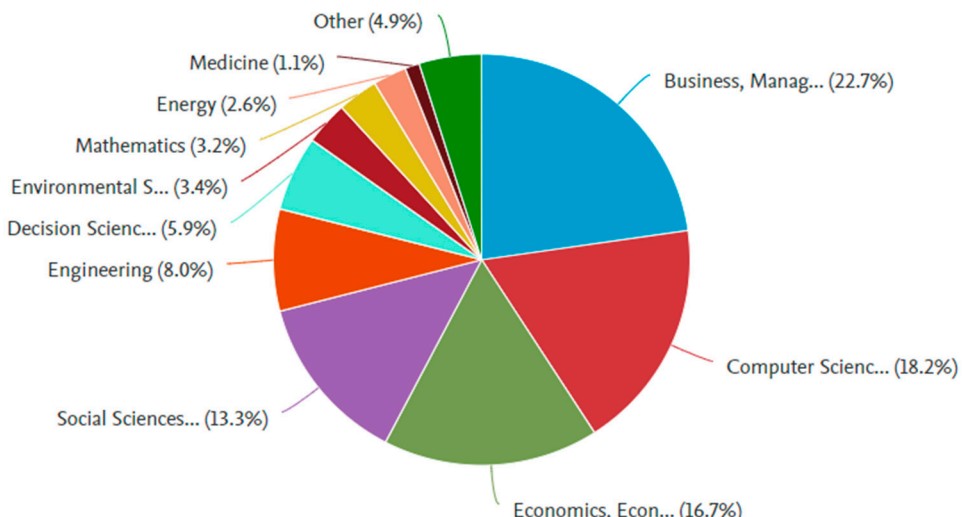

**Figure 9.** Distribution of disciplines for studies of FinTech. Note(s): This figure represents the distribution of disciplines in FinTech studies between 2015 and 2021. The data were taken from the Scopus database using the keyword "FinTech".

Figure 10 shows the betweenness centrality, which measures the number of times a node intersects the shortest path between two other nodes. This indicates an author's importance in connecting with other authors (Milian et al. 2019). The minimum number of citations in the figure is 10. Of the 60 sampled publications, 21 with links to each other were mapped, with a centrality between 0 and 10. Gomber et al. (2018), Shim and Shin (2016), and Schueffel (2016) have the highest centrality.

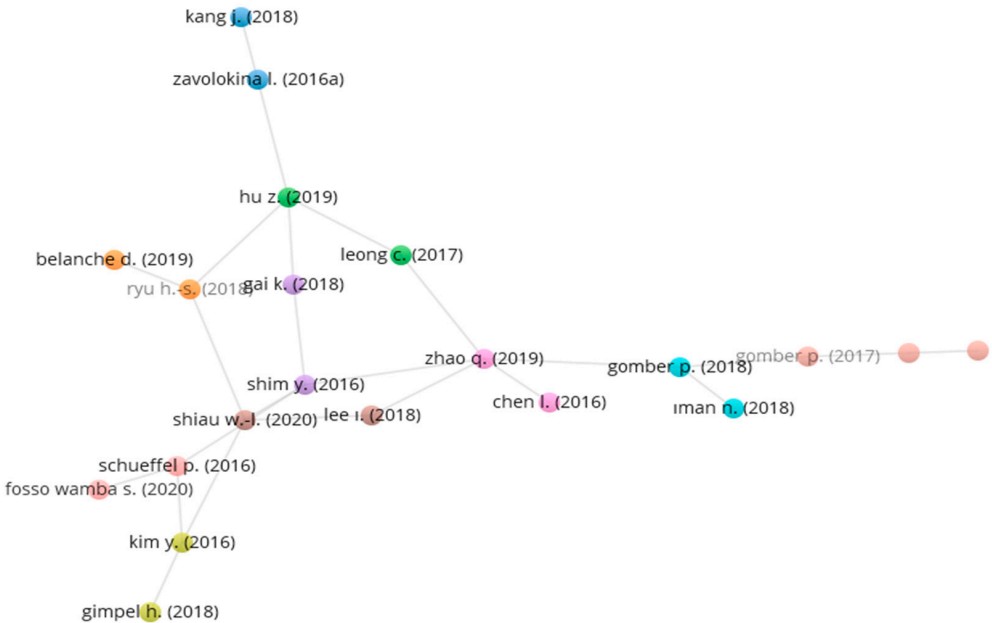

**Figure 10.** Intermediation as betweenness centrality. Note(s): This figure was created with a dataset from Scopus via VOSviewer.

### 4.4. Lotka's Law

Lotka's Law (Lotka 1926) predicts the number of publications published by each author in a particular field. That is, 60% of the authors will write one article, 15% will write two, 7% will write three, 4% will write four, etc. Figure 11 presents the results for papers on FinTech alongside the predicted distribution according to Lotka. It shows that 88.6% of authors have just one publication, 7.6% have two, and 2.5% have three. This indicates that FinTech authorship does not currently comply with Lotka's Law. The dashed line in the graph represents the graph that should be according to Lotka's Law.

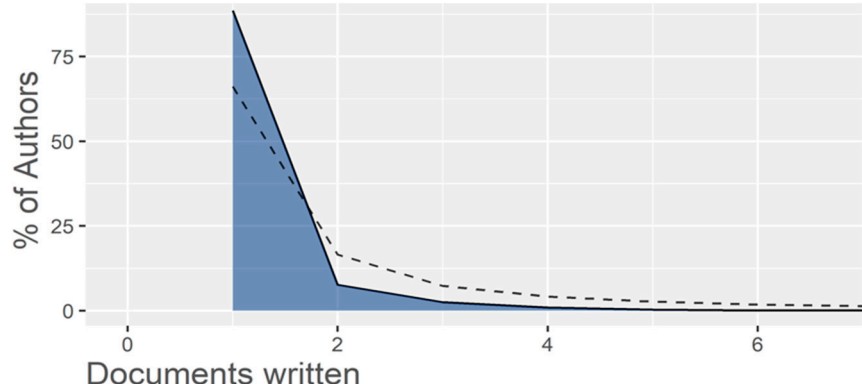

**Figure 11.** Lotka's Law of productivity, and actual authorship distribution. Note(s): This figure was created with a dataset from Scopus via R Studio.

### 4.5. Bradford's Law

Bradford's Law (Bradford [1929] 1985) measures whether journals have a core effect by dividing the journals in a specific field into three groups as outlined earlier. In the present study, 636 studies were published by 387 different journals and books. As Figure 12 shows, 40 journals and books accounted for 212 papers, 148 journals and books published 215 papers, and 199 journals and books published 209 articles. This suggests that FinTech research publishing is in line with Bradford's Law.

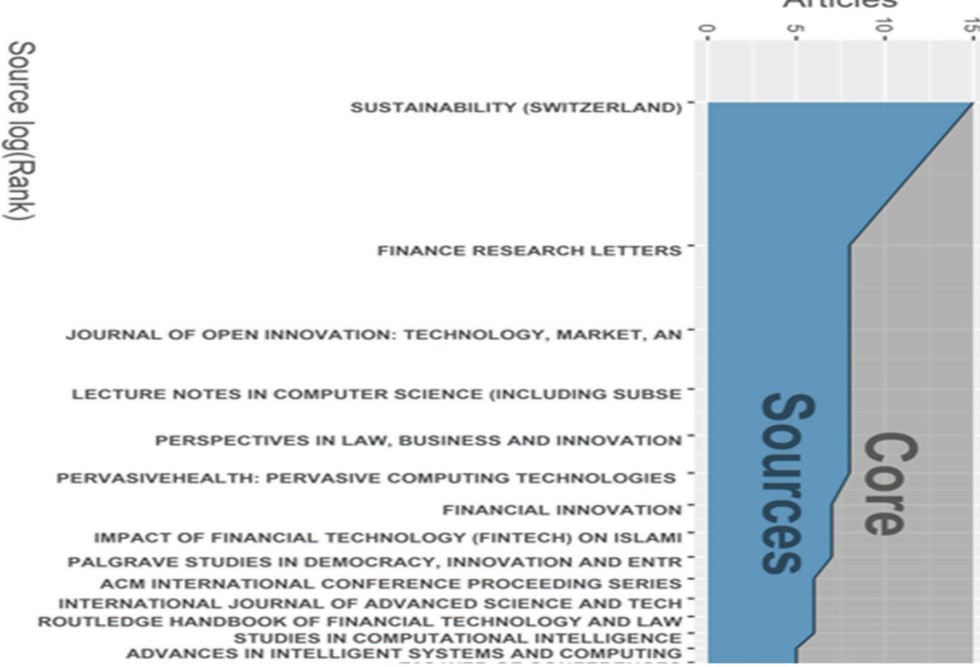

**Figure 12.** Bradford's Law of core publications and actual distribution by publications. Note(s): This figure was created with a dataset from Scopus via R Studio.

## 5. Results and Discussion

RQ1. How has the literature developed over time?

This question was answered by sub-questions RQ1.1, RQ1.2, and RQ1.3.

RQ1.1. What are the most influential studies and authors?

When the publications in 2018 are examined in Table 5, Gomber et al. (2018) with 47%, Lee and Shin (2018) with 41%, Buchak et al. (2018) with 60%, and Gai et al. (2018) with 35% received the most citations for 2021, while Gomber et al. (2017) with 37%, Gabor and Brooks (2017) with 40%, Haddad and Hornuf (2019) with 57%, and Shim and Shin (2016) with 30% received the most citations again in 2021. As the subject is still very new, researchers' interest is increasing continuously as indicated by the growing number of citations.

Gomber et al. (2018), Lee and Shin (2018), Gomber et al. (2017), and Gabor and Brooks (2017) all received at least 140 citations during the review period. Gomber et al. (2018) reported that the long-standing dominance of leading companies is at risk because they cannot effectively connect with the FinTech revolution. They presented a new FinTech-innovation-mapping approach that enables the assessment of the degree of changes and transformations in four financial services areas: operations management in financial services, technological innovations, multiple innovations, and issues related to investments. Lee and Shin (2018) examined FinTechs from a historical perspective and focused on various FinTech business models and investment types with their game-changing features. Gomber et al. (2017) introduced the institutions related to the digital finance cube, which includes three basic dimensions of digital finance and FinTech, related business functions,

applied technologies, and technological concepts. Gabor and Brooks (2017) examined the increasing importance of digital-based financial inclusion in the form of development interventions through FinTechs, government agencies, and other organizations. They concluded that FinTech-philanthropy development (FPD) creates ecosystems that map, expand, and monetize digital footprints. They also noted that the vision of the irrational client combines behavioral economics with predictive algorithms to accelerate access to finance and monitor adherence to them, while the digital revolution proposes new forms of profiling with financial(ized) inclusion that makes poor households new generators of financial assets.

RQ1.2. What are the main studies in FinTech?

As shown in Figure 6, Buchak et al. (2018) was one of the most influential works, followed by Gomber et al. (2018), Lee and Shin (2018), and Gomber et al. (2017). Buchak et al. (2018) studied how two forces, regulatory differences and technological advantages, contributed to this growth, due to the fact that shadow-bank market share in residential-mortgage origination nearly doubled from 2007 to 2015, with particularly dramatic growth among online "FinTech" lenders. Gai et al. (2018) surveyed FinTech by collecting and reviewing contemporary achievements that theoretically proposed a data-driven FinTech framework. The survey included five technical topics: security and privacy, data techniques, hardware and infrastructure, applications, and management and service models. They demonstrated the basics of creating active FinTech solutions. Schueffel (2016) offered a definition that is distinct as well as succinct in its communication, yet sufficiently broad in its range of application. Leong et al. (2017) examined the development of a FinTech company that gives micro-lending to university students in China. They showed how digital technology offers a firm strategic capability, how an alternative credit score can be calculated with unconventional data, and how financial coverage of market segments that are not previously covered can be realized. Haddad and Hornuf (2019) investigated the economic and technological determinants inducing entrepreneurs to establish ventures with the purpose of reinventing FinTech and found that the more difficult it is for companies to access loans, the higher is the number of FinTech startups in a country. Shim and Shin (2016) used Actor–Network Theory (ANT) to conduct a multi-level analysis of the historical development of China's FinTech industry as a stepping stone for investigating the interaction between it and the emerging social and political context. They also discussed policy implications of China's FinTech industry, focusing on the state's changing role in driving the growth of the national sector inside and outside.

RQ1.3. What are the distributions and effects of publications over time?

The sample included 636 studies focusing on FinTech applications, by 1445 different authors, from 387 different journals and books. The journals and books with the most publications (see Table 2) were as follows: Sustainability Switzerland with 15 publications, Perspectives in Law, Business and Innovation with 8 publications, and Impact of Financial Technology (Fintech) on Islamic Finance and Financial Stability with 7 publications. The top 10 journals and books include Industrial Management and Data Systems with 33.2 CPP, Financial Innovation with 15.28 CPP, Lecture Notes in Computer Science including subseries with 5 CPP, Journal of Open Innovation: Technology, Market, and Complexity with 4.12 CPP, and Finance Research Letters with 4 CPP. Thus, despite the novelty of this field, there are already many periodicals regularly publishing research on FinTech.

RQ2. What are the important topics in the FinTech literature?

Figure 9, which was developed according to Figure 8, showed the research disciplines of the sampled articles and the relationships between the most-frequently used words. The most common keywords in the papers were financial technology, blockchain, financial services, and financial inclusion. These keywords most often appeared in business-management sources (22.7%), followed by computer science (18.2%), economics (16.7%), and social science (13.3%). Thus, these four disciplines account for approximately 71% of

all publications on FinTech, which indicates that this field is currently confined to a few disciplines rather than being evenly dispersed across many.

From the analysis of the relationships between groups in the coding scheme, a framework has emerged for the literature summary, whose main axis is the FinTech activity sector, as shown in Figure 8. FinTech is most strongly connected to financial inclusion, China, and financial services, whereas blockchain has more connections with bitcoin, cryptocurrency, and smart contracts. Figures 8 and 9 formed the main backbone of the analysis for addressing RQ2.

RQ3. Are the results compatible with Lotka's Law?

Unsurprisingly, the vast majority of authors (88.6% of 1445) have just one publication, since FinTech has only recently entered the literature. Lotka's Law, however, predicts that only 60% of authors should have a single publication. Similarly, while 7.6% of authors examined had two publications, Lotka's Law predicts this should be around 15%. While just 2.5% of authors had three publications, Lotka's Law predicts 7%. Consequently, the distribution of authorship in FinTech does not conform to Lotka's Law.

RQ4. Are the results compatible with Bradford's Law?

Bradford's Law predicts that publications can be divided into three groups according to diminishing impact. The 40 journals that constitute the first group of publications in the study published 212 publications, 148 journals in the second group published 215, and 199 journals published 209 articles. The results of the study show that the first few journals published a third of the studies, followed by a large group that published the second third, and the largest number of journals published the remaining third. Thus, the distribution of publications by journals on FinTech is in line with Bradford's Law.

## 6. Conclusions

This study contributes to the understanding of the FinTech research phenomenon in five different ways in the scope of 636 publications obtained from Scopus between 2015 and 2021. First, FinTechs, which are increasingly influential globally, are also increasingly attracting attention in the scientific literature. Despite this growing interest, the research areas of publications on FinTechs are still not fully determined. The scarcity of mapping studies on FinTechs, as well as the lack of systematic reviews, suggests the need for a comprehensive analysis. The present study reveals the rapidly increasing interest in FinTech over the past six years as reflected in 636 publications from 387 journals and books predominantly representing four academic disciplines: business management, computer science, economics, and social science.

Second, this study identified the sub-topics and trends in publications on FinTechs along two axes. The first is financial services, financial inclusion, and financial technologies, where FinTech is centered. The access of investors and researchers to financial services, their involvement in financial business and transactions, and the use of financial technology are issues that have a significant impact on society. Research on the subject also shows that people of all levels are influenced by FinTech applications represented by these concepts and that traditional applications are quickly losing ground to FinTech applications. The second axis concerns the links to FinTech of cryptocurrency, bitcoin, and smart contracts, with blockchain as the hub. These new technological tools, in which information security is crucial, play an important role in making the individual and society freer. These technologies also demonstrate important security and privacy requirements that are needed in commercial life by opening the way for unmediated secure trade.

Thirdly, in order to make a complete definition of FinTech, this study investigated whether there is a consensus regarding the framework needed to describe FinTech. Research indicates the existence of a structure in which internet-based financial work and transactions can be conducted securely and privately, that facilitates access to information and finance, and that replaces the traditional financial structure with innovative companies.

Fourth, the study assessed the contributions and support of universities to academic research on FinTech. Universities in Asian countries receive more sponsorship and produce more articles, although their impact scores are lower. While the US and Europe have higher impact scores due to their current superiority in science and technology, Asian countries, especially China, are now focusing heavily on the issue and want to capture the trend of development in this area. The US leads in international cooperation between academics researching FinTech, followed by China, the United Kingdom, and Australia.

Fifth, while the distribution of authorship in this field conflicts with Lotka's Law, Bradford's Law was supported. The results of the study show that the first few journals published a third of the studies, followed by a large group that published the second third, and the largest number of journals published the remaining third. Given that FinTech is a very new field, it is possible that patterns of research publications will converge more with these laws in the future.

This study reflected the opinions and practices of all segments of FinTech research, as it included a wide range of articles, from traditional financial institutions to the FinTech ecosystem. As with similar studies adopting such a broad framework, however, the present study has limitations due to databases and search directories. The fact that databases such as Web of Science and ScienceDirect are not included in the study is a research limitation. In future studies, it is recommended to conduct comparative studies between Scopus, Web of Science, and ScienceDirect databases to expand the literature. The studies sampled here were also unique as they are some of the first in the field. This study examined research publications on FinTech based on four main research questions, which made it possible to deepen the study. FinTech research is predominantly conducted within business management, computer science, economics, and social science, thus paving the way for more in-depth research in these areas. In addition, several issues emerged that need to be examined more deeply, particularly FinTech's relationship with financial inclusion and financial services, and Blockchain's relationship with cryptocurrency and smart contracts. Examining these relationships to reveal their strength, causes, and effects would fill an important gap in the literature.

**Author Contributions:** Conceptualization, G.T. and U.B.G.; methodology, G.T.; software, G.T.; validation, G.T., U.B.G. and F.M.S.; formal analysis, G.T.; investigation, G.T., U.B.G. and F.M.S.; resources, G.T., U.B.G. and F.M.S.; data curation, G.T.; writing—original draft preparation, U.B.G.; writing—review and editing, G.T., U.B.G. and F.M.S.; visualization, G.T.; supervision, U.B.G.; project administration, G.T.; funding acquisition, F.M.S. All authors have read and agreed to the published version of the manuscript.

**Funding:** This research received no external funding.

**Institutional Review Board Statement:** Not applicable.

**Informed Consent Statement:** Not applicable.

**Data Availability Statement:** Data are contained within the article or available from referenced sources.

**Conflicts of Interest:** The authors declare no conflict of interest.

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
