# Peer review of "FinTech Companies: A Bibliometric Analysis†"

_ijfs, doi:10.3390/ijfs10010002_

Round 1

Reviewer 1 Report

Dear authors, 

You can see my comments in the attached file.

Faithfully!

Author Response

Thanks for your valuable contributions. Your comments have improved us as researchers as much as they have improved our article. What we learn while making these corrections will also help us in future articles.

In particular, expanding the data set until 2021 caused many changes in the article. The changes we made in line with your comments allowed us to make our article the best we could.

Response to Reviewer 1 Comments

  1. The purpose, the methods and methodology of the article are not defined in the abstract. It is necessary the authors to describe the findings and value added. In addition, it is essential the authors describe the “Research Questions” in the abstract.

Response: The purpose, methods, and methodology are added in the abstract. The findings and added value are added. The research questions were added to the abstract.

  1. It is significant the authors to portray the period of the research of the paper – in the abstract and in the conclusion.

Response: The period added to both abstract and conclusion

  1. In my opinion, RQ1.2. and RQ2 are overlapped. In this regard, they need to be formulated precisely.

Response: We consider that each research question is independent of the other. RQ.1.2 is a section that reveals which articles should be reviewed by an author who wants to research in the field. RQ. 2, on the other hand, is considered as a section that determines which disciplines are studied more and the direction of the trend.

  1. The data should be updated. The authors need to supplement 2019 and 2020 as a period in the research. The research period is not equal in the part discussion.

Response: In line with your valuable suggestions, the scope of the data set was expanded until 2021, and the data are analyzed, and major changes were included in the article.

  1. Line 28, please cite according to APA Direct Citation.

Response: Cited according to APA Direct Citation.

  1. Line 55-56, please provide page.

Response: Page provided.

  1. Line 103, please cite according to APA Direct Citation.

Response: Cited according to APA Direct Citation.

  1. Line 148, please specify which exactly is the “data set” – Fintech or Fintech and bibliometric analysis.

Response: Specified as bibliometric analysis.

  1. Line 154 and 155, please cite according to APA Direct Citation.

Response: Cited according to APA Direct Citation.

  1. Please, provide source at the end of the figures and tables.

Response: Sources provided at the end of figures and tables.

  1. Figure 12 – not readable. It will be appropriate the authors make summary of part 4.

Response: The figure has been made readable.

  1. Line 320, more formal language.

Response: Edited.

  1. From line 320 to Conclusion, please cite according to APA Direct Citation.

Response: Cited according to APA Direct Citation.

  1. After RQ4, the authors need to make summary.

Response: Summary added.

  1. Line 424, please explain what do you mean by 240 sources?

Response: Explained.

Reviewer 2 Report

Dear Author(s),
Please find below my recommendations and comments regarding your manuscript proposal entitled "FinTech Companies: A Bibliometric analysis".

1. First of all, the abstract must be rewritten. Here you must include the following important aspects:
- a short description of the concepts;
- the main challenge;
- your main findings and contributions.

2. In the Introduction section you are missing some valuable references for the scientific field of fintechs. Here I recommend you to cite the following valuable references:  https://doi.org/10.15240/tul/001/2021-2-007 (this article is about fintech services and factors determining the expected benefits of users), https://doi.org/10.1108/K-05-2020-0259 (this article is about social cryptocurrencies), https://doi.org/10.3390/sym11121449 (this article is about fintech 2.0 and 3.0 in the context of bank services), https://doi.org/10.1108/K-10-2020-0668 (this article is about blockchain and business intelligence), https://doi.org/10.15240/tul/001/2021-2-008 (this article is about financial performance and intangible assets).
Also, in the Introduction you have to clearly define and describe:
- the research question(s) (RQs). At this moment, you have some RQs in section 2.1, but they seems to be disconnected from the research gap because the research gap is not defined.
- the goal of the article;
- the research gap: please explain what you want to cover by your research, so that the readers know what is your contribution to the field of knowledge.

3. Before the section "2. Research Method", I recommend you to add some data about the market share of the main fintechs providers (Revolut, Monese etc). This way, the readers will understand the importance of this industry.

4. I didn't find any research hypothesis in your manuscript. Please define at least one research hypothesis, according to the modern requirements of the scientific articles.

5. You have 2 different section with the same number: 4.4. Please revise and correct this aspect.

6. Lotka's Law (section 4.4) should be tested through a statistical test (for example t-test or chi-square). The same remark for Bradford's Law.
At this moment, it is presented just an "intuitive" approach, but you should present it in a scientific manner.

7. Page 13, "RQ1.3. What are the distributions and effects of publications over time?": It is not usual to present ISSNs and ISBNs within an article body. Please reorganize this section by including the data into a relevant syntethic table.

8. Rows 407-408: "Consequently, the distribution of authorship in FinTech oes not conform to Lotka’s Law." Please correct the sequence "...in Fintech oes not..." and put "does" instead "oes".

9. In the Conclusions sections you must clearly present and describe the research limitations.

10. Based on your findings, the future research directions should be described in the Conclusions section.

Dear Author(s),
Please consider all the above remarks as being constructive recommendations in order to improve the general quality of your manuscript proposal.

Kind Regards!

Author Response

Thanks for your valuable contributions. Your comments have improved us as researchers as much as they have improved our article. What we learn while making these corrections will also help us in future articles.

In particular, expanding the data set until 2021 caused many changes in the article. The changes we made in line with your comments allowed us to make our article the best we could.

Response to Reviewer 2 Comments

. First of all, the abstract must be rewritten. Here you must include the following important aspects:

- a short description of the concepts;

- the main challenge;

- your main findings and contributions.

Response: The purpose, methods, and methodology are added in the abstract. Description of the concepts, main challenge findings, and contributions are added. The research questions were added to the abstract.

  1. In the Introduction section you are missing some valuable references for the scientific field of fintechs. Here I recommend you to cite the following valuable references: https://doi.org/10.15240/tul/001/2021-2-007 (this article is about fintech services and factors determining the expected benefits of users), https://doi.org/10.1108/K-05-2020-0259 (this article is about social cryptocurrencies), https://doi.org/10.3390/sym11121449 (this article is about fintech 2.0 and 3.0 in the context of bank services), https://doi.org/10.1108/K-10-2020-0668 (this article is about blockchain and business intelligence), https://doi.org/10.15240/tul/001/2021-2-008 (this article is about financial performance and intangible assets).

Also, in the Introduction you have to clearly define and describe:

- the research question(s) (RQs). At this moment, you have some RQs in section 2.1, but they seems to be disconnected from the research gap because the research gap is not defined.

- the goal of the article;

- the research gap: please explain what you want to cover by your research, so that the readers know what is your contribution to the field of knowledge.

Response: Thank you for your recommendations. All recommended studies were added except the last one. In addition, the goal of the article was expressed more clearly and added to the introduction.

  1. Before the section "2. Research Method", I recommend you to add some data about the market share of the main fintechs providers (Revolut, Monese etc). This way, the readers will understand the importance of this industry.

Response: Thanks for your contribution. It would be nice to have more examples about the market share of the main FinTech providers and make readers understand the importance of this industry. However, we did not do this due to concerns that the introduction would be too long to steal the light from the bibliometric analysis, which is the original part of the article. We will focus more on the market share of the main FinTech providers in our future work.

  1. I didn't find any research hypothesis in your manuscript. Please define at least one research hypothesis, according to the modern requirements of the scientific articles.

Response: The bibliometric analysis aims to present a study that guides the authors who want to publish in the related field. Studies using bibliometric analysis techniques classify the complex structure of the literature and draw a roadmap for the researcher. For this reason, it is not common to include hypotheses in articles about bibliometric analysis. However, as we agree with your valuable comment, we have added Research Questions that are not available in other bibliometric analysis studies.

  1. You have 2 different section with the same number: 4.4. Please revise and correct this aspect.

Response: Corrected.

  1. Lotka's Law (section 4.4) should be tested through a statistical test (for example t-test or chi-square). The same remark for Bradford's Law. At this moment, it is presented just an "intuitive" approach, but you should present it in a scientific manner.

Response: The analyzes in Lotka's Law and Bradford's Law section are the results that were coded by the authors in the R Studio program through formulas taken from the original articles. Formulas contain their own statistical accuracy. For this reason, statistical tests such as t-test and chi-square cannot be applied to related fields.

  1. Page 13, "RQ1.3. What are the distributions and effects of publications over time?": It is not usual to present ISSNs and ISBNs within an article body. Please reorganize this section by including the data into a relevant syntethic table.
    Response:
    Corrected.
  2. Rows 407-408: "Consequently, the distribution of authorship in FinTech oes not conform to Lotka’s Law." Please correct the sequence "...in Fintech oes not..." and put "does" instead "oes".

Response: Corrected.

  1. In the Conclusions sections you must clearly present and describe the research limitations.

Response: Added. (As with similar studies adopting such a broad framework, however, the present study has limitations due to databases and search directories. The fact that databases such as Web of Science and ScienceDirect are not included in the study is a research limitation.)

  1. Based on your findings, the future research directions should be described in the Conclusions section.

Response: Added. (In addition, several issues emerged that need to be examined more deeply, particularly FinTech’s relationship with financial inclusion and financial services, and Blockchain’s relationship with cryptocurrency and smart contracts. Examining these relationships to reveal their strength, causes, and effects would fill an important gap in the literature.)

Reviewer 3 Report

Dear Authors,

Thank you so much for the opportunity to review your manuscript, the topic of which is truly interesting to me!

You can find my detailed suggestions and comments in the attached file.

Looking forward to receiving your reply!

Author Response

Thanks for your valuable contributions. Your comments have improved us as researchers as much as they have improved our article. What we learn while making these corrections will also help us in future articles.

In particular, expanding the data set until 2021 caused many changes in the article. The changes we made in line with your comments allowed us to make our article the best we could.

Response to Reviewer 3 Comments

  1. The aim/goal/purpose of the article should be precisely formulates in the Abstract. Moreover, the studied period, during which the database for the research has been established must be indicated in the Abstract as well. Hence, the methods and methodology applied must be mentioned together with what did the authors established by the research as findings and value added.

Response: The purpose, methods, and methodology are added in the abstract. The findings and added value are added. The research questions were added to the abstract. The period was added to the abstract.

  1. The authors should revise the manuscript and be more precise when citing directly, according to the APA style. Examples: line 28, line 55, line 103, line 154-155, line 320-323, etc.

Response: Cited according to APA Direct Citation.

  1. The authors have to update the literature review on the studied topic – Fintech with more recent research data. I am more than certain that from 2018 to percent day the scientific community have developed quite a significant number of studies on the topic.

Response: In line with your valuable suggestions, the scope of the data set was expanded until 2021, and the data are analyzed, and major changes were included in the article.

  1. Line 130 – once again, update the research period and be more precise – “over the time” is a quite a vague term.

Response: Updated.

  1. Line 144-145 – the statement must be revised, not clear what the authors mean, is their research “detailed and comprehensive” or the bibliometric analysis in general is “detailed and comprehensive”? Furthermore, in Scopus and in Web of Science one can find at least 10 research papers concerning the bibliometric analysis on Fintech for the period 2018-2021.

Response: Revised and updated.

  1. The authors have to revise the manuscript and apply more formal language such as: line 153, line 186, line 191, line 320, etc.

Response: Revised.

  1. All figures and tables must have a Source!

Response: Added resources to all figures and tables. Original tables and figures created by the researcher were shown as notes.

  1. Not clear what do the authors mean by source in the statement “401 studies were published by 240 different sources”?

Response: Revised.

  1. Overall, the paper lacks scientific soundness due to the fact that the research period ended 4 years ago. The authors need to provide further data, at least till the end of 2020. One further, starting from the literature review, I believe the authors have to explore the development of the Fintech sector together with the influence that the current pandemic has on it. Many scientist have elaborated abundant quantity of research materials on the topic since 2018.

Response: As you suggested, the dataset has been updated to include developments in recent years.

Reviewer 4 Report

Dear author, 

You have very good work on this paper. I believe you just must take into account some notes, and some of them are simple in form.

In citations, you always use "(author, year)". But it depends on the phrase. Sometimes you should use "author (year)". Have a look at lines 28, 128, 154, 255, and so on. Also in tables: table 5.

Also, even though the tables' acronyms are already said (line 197), in each table you should repeat them in the legend. See, for example, tables 2, 3, and 4.

Sources on tables and figures are essential. Even if it is elaborated by you, this must be mentioned. Tables in the paper do not have a source, and figures from 2 onwards also do not.

Line 182 seems more like findings, and results should be in line 315. Titles shall mention what is in the text.

Figure 12 is not well presented. Can you find a better way to present the information?

It also is not clear the interest of Lotkas's and Bradford Laws. In my opinion, you should explain better why this analysis or leave it.

At last, you missed a "d" in line 408 and have an extra point (.) in line 440.

I´m hoping to see your article published.

Author Response

Thanks for your valuable contributions. Your comments have improved us as researchers as much as they have improved our article. What we learn while making these corrections will also help us in future articles.

In particular, expanding the data set until 2021 caused many changes in the article. The changes we made in line with your comments allowed us to make our article the best we could.

Response to Reviewer 4 Comments

You have very good work on this paper. I believe you just must take into account some notes, and some of them are simple in form.

Response: Thank you for your encouraging comment.

In citations, you always use "(author, year)". But it depends on the phrase. Sometimes you should use "author (year)". Have a look at lines 28, 128, 154, 255, and so on. Also in tables: table 5.

Response: Cited according to APA Direct Citation.

Also, even though the tables' acronyms are already said (line 197), in each table you should repeat them in the legend. See, for example, tables 2, 3, and 4.

Response: Revised. 

Sources on tables and figures are essential. Even if it is elaborated by you, this must be mentioned. Tables in the paper do not have a source, and figures from 2 onwards also do not.

Response: Added resources to all figures and tables. Original tables and figures created by the researcher were shown as notes.

Line 182 seems more like findings, and results should be in line 315. Titles shall mention what is in the text.

Response: Revised. 

Figure 12 is not well presented. Can you find a better way to present the information?

Response: The figure has been made readable.

It also is not clear the interest of Lotkas's and Bradford Laws. In my opinion, you should explain better why this analysis or leave it.

Response: Revised. We did our best to be clearer.

At last, you missed a "d" in line 408 and have an extra point (.) in line 440.

Response: Corrected.

Round 2

Reviewer 1 Report

Dear authors, thank you for your comments regarding my remarks! 

Author Response

Dear reviewer, thanks to your valuable contributions, our article has reached its best form. Thank you!

Reviewer 2 Report

Dear Author(s),

I have read the new version of the manuscript and I consider you addressed my remarks from the previous round of review.

Regarding the present form of the manuscript, I have only some minor remarks:

  1. The last reference from the list of references (page 19, rows 667-668) is not numbered. Please revise and correct this issue.
  2. I recommend you to complete the section containing the future directions from the Conclusion section (rows 503-507) with a new future direction: a comparative study between Scopus, Web of Science and ScienceDirect. My recommendation is based on one of your research limitation (the data used in your research comes only from Scopus database).

Kind Regards!

Author Response

Dear reviewer, changes have been made according to your suggestions. Thanks to your valuable contributions, our article has reached its best form. Thank you!

Reviewer 3 Report

Dear Authors,

Thank you very much for taking into account my comments and recommendations in the revised version of your manuscript!

Although, I believe there is still some further points to take into account in order to increase the scientific soundness of your research. Namely, as evident from the first part of the manuscript, you have updated the research scope period, but the second part - represented in Figure 6 and Figure 10 such updated is missing. You could you consider providing it.

Moreover, if possible make the data in Figure 9 more readable.

Hope my remarks are helpful!

Sincerely!

Author Response

Dear reviewer, after changing the scope period in our data set, we did all the analysis from the beginning. The reason why we cannot see more recent articles in Figure 6 and Figure 10 is because the analysis creates links according to citation weights. Since the number of citations of older articles is higher, the results of the analysis are like this.
We tried to make Figure 9 more readable.

Upon your request, we can share our data set, old and new analysis results with you.

Thanks to your valuable contributions, our article has reached its best form. Thank you!

This manuscript is a resubmission of an earlier submission. The following is a list of the peer review reports and author responses from that submission.